# Fast, Sample-Efficient Algorithms for Structured Phase Retrieval

**Gauri jagatap**
Electrical and Computer Engineering
Iowa State University
gauri@iastate.edu

**Chinmay Hegde**
Electrical and Computer Engineering
Iowa State University
chinmay@iastate.edu

## Abstract

We consider the problem of recovering a signal $\mathbf{x}^* \in \mathbb{R}^n$, from magnitude-only measurements, $y_i = |\langle \mathbf{a}_i, \mathbf{x}^* \rangle|$ for $i = \{1, 2, \ldots, m\}$. Also known as the *phase retrieval problem*, it is a fundamental challenge in nano-, bio- and astronomical imaging systems, and speech processing. The problem is ill-posed, and therefore additional assumptions on the signal and/or the measurements are necessary.

In this paper, we first study the case where the underlying signal $\mathbf{x}^*$ is $s$-sparse. We develop a novel recovery algorithm that we call *Compressive Phase Retrieval with Alternating Minimization*, or *CoPRAM*. Our algorithm is simple and can be obtained via a natural combination of the classical alternating minimization approach for phase retrieval, with the CoSaMP algorithm for sparse recovery. Despite its simplicity, we prove that our algorithm achieves a sample complexity of $\mathcal{O}\left(s^2 \log n\right)$ with Gaussian samples, which matches the best known existing results. It also demonstrates linear convergence in theory and practice and requires no extra tuning parameters other than the signal sparsity level $s$.

We then consider the case where the underlying signal $\mathbf{x}^*$ arises from *structured* sparsity models. We specifically examine the case of *block-sparse* signals with uniform block size of $b$ and block sparsity $k = s/b$. For this problem, we design a recovery algorithm that we call *Block CoPRAM* that further reduces the sample complexity to $\mathcal{O}\left(ks \log n\right)$. For sufficiently large block lengths of $b = \Theta(s)$, this bound equates to $\mathcal{O}\left(s \log n\right)$. To our knowledge, this constitutes the first end-to-end linearly convergent family of algorithms for phase retrieval where the Gaussian sample complexity has a sub-quadratic dependence on the sparsity level of the signal.

## 1 Introduction

### 1.1 Motivation

In this paper, we consider the problem of recovering a signal $\mathbf{x}^* \in \mathbb{R}^n$ from (possibly noisy) *magnitude-only* linear measurements. That is, for sampling vector $\mathbf{a}_i \in \mathbb{R}^n$, if

$$y_i = |\langle \mathbf{a}_i, \mathbf{x}^* \rangle|, \quad \text{for } i = 1, \ldots, m, \tag{1}$$

then the task is to recover $\mathbf{x}^*$ using the measurements $\mathbf{y}$ and the sampling matrix $\mathbf{A} = [\mathbf{a}_1 \ldots \mathbf{a}_m]^\top$.

Problems of this kind arise in numerous scenarios in machine learning, imaging, and statistics. For example, the classical problem of *phase retrieval* is encountered in imaging systems such as diffraction imaging, X-ray crystallography, ptychography, and astronomy [1, 2, 3, 4, 5]. For such imaging systems, the optical sensors used for light acquisition can only record the intensity of the light waves but not their phase. In terms of our setup, the vector $\mathbf{x}^*$ corresponds to an image (with a resolution of $n$ pixels) and the measurements correspond to the magnitudes of its 2D Fourier coefficients. The goal is to stably recover the image $\mathbf{x}^*$ using as few observations $m$ as possible.

Despite the prevalence of several heuristic approaches [6, 7, 8, 9], it is generally accepted that (1) is a challenging nonlinear, ill-posed inverse problem in theory and practice. For generic $\mathbf{a}_i$ and $\mathbf{x}^*$, one can show that (1) is *NP-hard* by reduction from well-known combinatorial problems [10]. Therefore, additional assumptions on the signal $\mathbf{x}^*$ and/or the measurement vectors $\mathbf{a}_i$ are necessary.

A recent line of breakthrough results [11, 12] have provided efficient algorithms for the case where the measurement vectors arise from certain multi-variate probability distributions. The seminal paper by Netrapalli *et al.* [13] provides the first rigorous justification of classical heuristics for phase retrieval based on alternating minimization. However, all these newer results require an "overcomplete" set of observations, i.e., the number of observations $m$ exceeds the problem dimension $n$ ($m = \mathcal{O}(n)$ being the tightest evaluation of this bound [14]). This requirement can pose severe limitations on computation and storage, particularly when $m$ and $n$ are very large.

One way to mitigate the dimensionality issue is to use the fact that in practical applications, $\mathbf{x}^*$ often obeys certain *low-dimensional* structural assumptions. For example, in imaging applications $\mathbf{x}^*$ is *s-sparse* in some known basis, such as identity or wavelet. For transparency, we assume the canonical basis for sparsity throughout this paper. Similar structural assumptions form the core of sparse recovery, and streaming algorithms [15, 16, 17], and it has been established that only $\mathcal{O}\left(s \log \frac{n}{s}\right)$ samples are necessary for stable recovery of $\mathbf{x}^*$, which is information-theoretically optimal [18].

Several approaches for solving the sparsity-constrained version of (1) have been proposed, including alternating minimization [13], methods based on convex relaxation [19, 20, 21], and iterative thresholding [22, 23]. Curiously, all of the above techniques incur a sample complexity of $\Omega(s^2 \log n)$ for stable recovery, which is quadratically worse than the information-theoretic limit [18] of $\mathcal{O}\left(s \log \frac{n}{s}\right)$[1]. Moreover, most of these algorithms have quadratic (or worse) running time [19, 22], stringent assumptions on the nonzero signal coefficients [13, 23], and require several tuning parameters [22, 23].

Finally, for specific applications, more refined structural assumptions on $\mathbf{x}^*$ are applicable. For example, point sources in astronomical images often produce *clusters* of nonzero pixels in a given image, while wavelet coefficients of natural images often can be organized as connected *sub-trees*. Algorithms that leverage such *structured sparsity* assumptions have been shown to achieve considerably improved sample-complexity in statistical learning and sparse recovery problems using block-sparsity [30, 31, 32, 33], tree sparsity [34, 30, 35, 36], clusters [37, 31, 38], and graph models [39, 38, 40]. However, these models have not been understood in the context of phase retrieval.

## 1.2 Our contributions

The contributions in this paper are two-fold. First, we provide a new, flexible algorithm for sparse phase retrieval that matches state of the art methods both from a statistical as well as computational viewpoint. Next, we show that it is possible to extend this algorithm to the case where the signal is block-sparse, thereby *further* lowering the sample complexity of stable recovery. Our work can be viewed as a first step towards a general framework for phase retrieval of structured signals from Gaussian samples.

***Sparse phase retrieval.*** We first study the case where the underlying signal $\mathbf{x}^*$ is *s*-sparse. We develop a novel recovery algorithm that we call *Compressive Phase Retrieval with Alternating Minimization*, or *CoPRAM*[2]. Our algorithm is simple and can be obtained via a natural combination of the classical alternating minimization approach for phase retrieval with the CoSaMP [41] algorithm for sparse recovery (CoSAMP also naturally extends to several sparsity models [30]). We prove that our algorithm achieves a sample complexity of $\mathcal{O}(s^2 \log n)$ with Gaussian measurement vectors $\mathbf{a}_i$ in order to achieve *linear convergence*, matching the best among all existing results. An appealing feature of our algorithm is that it requires no extra *a priori* information other than the signal sparsity level *s*, and no assumptions on the nonzero signal coefficients. To our knowledge, this is the first algorithm for sparse phase retrieval that simultaneously achieves all of the above properties. We use CoPRAM as the basis to formulate a block-sparse extension (Block CoPRAM).

***Block-sparse phase retrieval.*** We consider the case where the underlying signal $\mathbf{x}^*$ arises from *structured* sparsity models, specifically *block-sparse* signals with uniform block size $b$ (i.e., *s* nonzeros equally grouped into $k = s/b$ blocks). For this problem, we design a recovery algorithm that we

Table 1: *Comparison of (Gaussian sample) sparse phase retrieval algorithms. Here, $n, s, k = s/b$ denote signal length, sparsity, and block-sparsity. $\mathcal{O}_\epsilon(\cdot)$ hides polylogarithmic dependence on $\frac{1}{\epsilon}$.*

| Algorithm | Sample complexity | Running time | Assumptions | Parameters |
|---|---|---|---|---|
| AltMinSparse | $\mathcal{O}_\epsilon\left(s^2 \log n + s^2 \log^3 s\right)$ | $\mathcal{O}_\epsilon\left(s^2 n \log n\right)$ | $x^*_{\min} \approx \frac{c}{\sqrt{s}}\|\mathbf{x}^*\|_2$ | none |
| $\ell_1$-PhaseLift | $\mathcal{O}\left(s^2 \log n\right)$ | $\mathcal{O}\left(\frac{n^3}{\epsilon^2}\right)$ | none | none |
| Thresholded WF | $\mathcal{O}\left(s^2 \log n\right)$ | $\mathcal{O}_\epsilon\left(n^2 \log n\right)$ | none | step $\mu$, thresholds $\alpha, \beta$ |
| SPARTA | $\mathcal{O}\left(s^2 \log n\right)$ | $\mathcal{O}_\epsilon\left(s^2 n \log n\right)$ | $x^*_{\min} \approx \frac{c}{\sqrt{s}}\|\mathbf{x}^*\|_2$ | step $\mu$, threshold $\gamma$ |
| CoPRAM | $\mathcal{O}\left(s^2 \log n\right)$ | $\mathcal{O}_\epsilon\left(s^2 n \log n\right)$ | none | none |
| Block CoPRAM | $\mathcal{O}\left(ks \log n\right)$ | $\mathcal{O}_\epsilon\left(ksn \log n\right)$ | none | none |

call *Block CoPRAM*. We analyze this algorithm and show that leveraging block-structure reduces the sample complexity for stable recovery to $\mathcal{O}(ks \log n)$. For sufficiently large block lengths $b = \Theta(s)$, this bound equates to $\mathcal{O}(s \log n)$. To our knowledge, this constitutes the first phase retrieval algorithm where the Gaussian sample complexity has a sub-quadratic dependence on the sparsity $s$ of the signal.

A comparative description of the performance of our algorithms is presented in Table 1.

## 1.3 Techniques

***Sparse phase retrieval.*** Our proposed algorithm, CoPRAM, is conceptually very simple. It integrates existing approaches in stable sparse recovery (specifically, the CoSaMP algorithm [41]) with the alternating minimization approach for phase retrieval proposed in [13].

A similar integration of sparse recovery with alternating minimization was also introduced in [13]; however, their approach only succeeds when the true support of the underlying signal is accurately identified during initialization, which can be unrealistic. Instead, CoPRAM permits the support of the estimate to evolve across iterations, and therefore can iteratively "correct" for any errors made during the initialization. Moreover, their analysis requires using fresh samples for every new update of the estimate, while ours succeeds in the (more practical) setting of using all the available samples.

Our first challenge is to identify a good initial guess of the signal. As is the case with most non-convex techniques, CoPRAM requires an initial estimate $\mathbf{x}^0$ that is close to the true signal $\mathbf{x}^*$. The basic idea is to identify "important" co-ordinates by constructing suitable biased estimators of each signal coefficient, followed by a specific eigendecomposition. The initialization in CoPRAM is far simpler than the approaches in [22, 23]; requiring no pre-processing of the measurements and or tuning parameters other than the sparsity level $s$. A drawback of the theoretical results of [23] is that they impose a requirement on signal coefficients: $\min_{j \in S} |x^*_j| = C \|\mathbf{x}^*\|_2 / \sqrt{s}$. However, this assumption disobeys the power-law decay observed in real world signals. Our approach also differs from [22], where they estimate an initial support based on a parameter-dependent threshold value. Our analysis removes these requirements; we show that a coarse estimate of the support, coupled with the spectral technique in [22, 23] gives us a suitable initialization. A sample complexity of $\mathcal{O}\left(s^2 \log n\right)$ is incurred for achieving this estimate, matching the best available previous methods.

Our next challenge is to show that given a good initial guess, alternatingly estimating the phases and non-zero coefficients (using CoSaMP) gives a rapid convergence to the desired solution. To this end, we use the analysis of CoSaMP [41] and leverage a recent result by [42], to show per step decrease in the signal estimation error using the generic chaining technique of [43, 44]. In particular, we show that any "phase errors" made in the initialization, can be suitably controlled across different estimates.

***Block-sparse phase retrieval.*** We use CoPRAM to establish its extension Block CoPRAM, which is a novel phase retrieval strategy for block sparse signals from Gaussian measurements. Again, the algorithm is based on a suitable initialization followed by an alternating minimization procedure, mirroring the steps in CoPRAM. To our knowledge, this is the first result for phase retrieval under more refined structured sparsity assumptions on the signal.

As above, the first stage consists of identifying a good initial guess of the solution. We proceed as in CoPRAM, isolating *blocks* of nonzero coordinates, by constructing a biased estimator for the "mass" of each block. We prove that a good initialization can be achieved using this procedure using only $\mathcal{O}(ks \log n)$ measurements. When the block-size is large enough ($b = \Theta(s)$), the sample complexity of the initialization is *sub-quadratic* in the sparsity level $s$ and only a logarithmic factor above the

information-theoretic limit $\mathcal{O}(s)$ [30]. In the second stage, we demonstrate a rapid descent to the desired solution. To this end, we replace the CoSaMP sub-routine in CoPRAM with the *model-based CoSaMP* algorithm of [30], specialized to block-sparse recovery. The analysis proceeds analogously as above. To our knowledge, this constitutes the first end-to-end algorithm for phase retrieval (from Gaussian samples) that demonstrates a sub-quadratic dependence on the sparsity level of the signal.

## 1.4 Prior work

The phase retrieval problem has received significant attention in the past few years. Convex methodologies to solve the problem in the *lifted* framework include *PhaseLift* and its variations [11, 45, 46, 47]. Most of these approaches suffer severely in terms of computational complexity. *PhaseMax*, produces a convex relaxation of the phase retrieval problem similar to basis pursuit [48]; however it is not empirically competitive. Non-convex algorithms typically rely on finding a good initial point, followed by minimizing a quadratic (Wirtinger Flow [12, 14, 49]) or moduli ( [50, 51]) measurement loss function. Arbitrary initializations have been studied in a polynomial-time trust-region setting in [52].

Some of the convex approaches in sparse phase retrieval include [19, 53], which uses a combination of trace-norm and $\ell$-norm relaxation.Constrained sensing vectors have been used [25] at optimal sample complexity $\mathcal{O}\left(s \log \frac{n}{s}\right)$. Fourier measurements have been studied extensively in the convex [54] and non-convex [55] settings. More non-convex approaches for sparse phase retrieval include [13, 23, 22] which achieve Gaussian sample complexities of $\mathcal{O}\left(s^2 \log n\right)$.

Structured sparsity models such as groups, blocks, clusters, and trees can be used to model real-world signals.Applications of such models have been developed for sparse recovery [30, 33, 39, 38, 40, 56, 34, 35, 36] as well as in high-dimensional optimization and statistical learning [32, 31]. However, to the best of our knowledge, there have been no rigorous results that explore the impact of structured sparsity models for the phase retrieval problem.

## 2  Paper organization and notation

The remainder of the paper is organized as follows. In Sections 3 and 4, we introduce the CoPRAM and Block CoPRAM algorithms respectively, and provide a theoretical analysis of their statistical performance. In Section 5 we present numerical experiments for our algorithms.

Standard notation for matrices (capital, bold: $\mathbf{A}, \mathbf{P}$, etc.), vectors (small, bold: $\mathbf{x}, \mathbf{y}$, etc.) and scalars ( $\alpha, c$ etc.) hold. Matrix and vector transposes are represented using $\top$ (eg. $\mathbf{x}^\top$ and $\mathbf{A}^\top$) respectively. The diagonal matrix form of a column vector $\mathbf{y} \in \mathbb{R}^m$ is represented as $\mathrm{diag}(\mathbf{y}) \in \mathbb{R}^{m \times m}$. Operator $\mathrm{card}(S)$ represents cardinality of $S$. Elements of $\mathbf{a}$ are distributed according to the zero-mean standard normal distribution $\mathcal{N}(0,1)$. The phase is denoted using $\mathrm{sign}(\mathbf{y}) \equiv \mathbf{y}/|\mathbf{y}|$ for $\mathbf{y} \in \mathbb{R}^m$, and $\mathrm{dist}(\mathbf{x}_1, \mathbf{x}_2) \equiv \min(\|\mathbf{x}_1 - \mathbf{x}_2\|_2, \|\mathbf{x}_1 + \mathbf{x}_2\|_2)$ for every $\mathbf{x}_1, \mathbf{x}_2 \in \mathbb{R}^n$ is used to denote "distance", upto a global phase factor (both $\mathbf{x} = \mathbf{x}^*, -\mathbf{x}^*$ satisfy $\mathbf{y} = |\mathbf{A}\mathbf{x}|$). The projection of vector $\mathbf{x} \in \mathbb{R}^n$ onto a set of coordinates $S$ is represented as $\mathbf{x}_S \in \mathbb{R}^n$, $x_{Sj} = x_j$ for $j \in S$, and $0$ elsewhere. Projection of matrix $\mathbf{M} \in \mathbb{R}^{n \times n}$ onto $S$ is $\mathbf{M}_S \in \mathbb{R}^{n \times n}$, $M_{Sij} = M_{ij}$ for $i, j \in S$, and $0$ elsewhere. For faster algorithmic implementations, $\mathbf{M}_S$ can be assumed to be a truncated matrix $\mathbf{M}_S \in \mathbb{R}^{s \times s}$, discarding all row and column elements corresponding to $S^c$. The element-wise inner product of two vectors $\mathbf{y}_1$ and $\mathbf{y}_2 \in \mathbb{R}^m$ is represented as $\mathbf{y}_1 \circ \mathbf{y}_2$. Unspecified large and small constants are represented by $C$ and $\delta$ respectively. The abbreviation *w.h.p.* denotes "with high probability".

## 3  Compressive phase retrieval

In this section, we propose a new algorithm for solving the sparse phase retrieval problem and analyze its performance. Later, we will show how to extend this algorithm to the case of more refined structural assumptions about the underlying sparse signal.

We first provide a brief outline of our proposed algorithm. It is clear that the *sparse* recovery version of (1) is highly non-convex, and possibly has multiple local minima[22]. Therefore, as is typical in modern non-convex methods [13, 23, 57] we use an spectral technique to obtain a good initial estimate. Our technique is a modification of the initialization stages in [22, 23], but requires no tuning parameters or assumptions on signal coefficients, except for the sparsity $s$. Once an appropriate initial

---

**Algorithm 1** CoPRAM: Initialization.

---

**input** $\mathbf{A}, \mathbf{y}, s$.
    Compute signal power: $\phi^2 = \frac{1}{m} \sum_{i=1}^{m} y_i^2$.
    Compute signal marginals: $M_{jj} = \frac{1}{m} \sum_{i=1}^{m} y_i^2 a_{ij}^2 \quad \forall j$.
    Set $\hat{S} \leftarrow j$'s corresponding to top-$s$ $M_{jj}$'s.
    Set $\mathbf{v}_1 \leftarrow$ top singular vector of $\mathbf{M}_{\hat{S}} = \frac{1}{m} \sum_{i=1}^{m} y_i^2 \mathbf{a}_{i\hat{S}} \mathbf{a}_{i\hat{S}}^{\top} \quad \in \mathbb{R}^{s \times s}$.
    Compute $\mathbf{x^0} \leftarrow \phi \mathbf{v}$, where $\mathbf{v} \leftarrow \mathbf{v}_1$ for $\hat{S}$ and $\mathbf{0} \in \mathbb{R}^{n-s}$ for $\hat{S}^c$.
**output** $\mathbf{x^0}$.

---

**Algorithm 2** CoPRAM: Descent.

---

**input** $\mathbf{A}, \mathbf{y}, \mathbf{x^0}, s, t_0$.
    Initialize $\mathbf{x^0}$ according to Algorithm 1.
    **for** $t = 0, \cdots, t_0 - 1$ **do**
      $\mathbf{P}^{t+1} \leftarrow \operatorname{diag}\left(\operatorname{sign}\left(\mathbf{A}\mathbf{x}^t\right)\right)$,
      $\mathbf{x}^{t+1} \leftarrow \text{COSAMP}(\frac{1}{\sqrt{m}}\mathbf{A}, \frac{1}{\sqrt{m}}\mathbf{P}^{t+1}\mathbf{y}, s, \mathbf{x}^t)$.
    **end for**
**output** $\mathbf{z} \leftarrow \mathbf{x}^{t_0}$.

---

estimate is chosen, we then show that a simple alternating-minimization algorithm, based on the algorithm in [13] will converge rapidly to the underlying true signal. We call our overall algorithm *Compressive Phase Retrieval with Alternating Minimization* (CoPRAM) which is divided into two stages: *Initialization* (Algorithm 1) and *Descent* (Algorithm 2).

## 3.1 Initialization

The high level idea of the first stage of CoPRAM is as follows; we use measurements $y_i$ to construct a *biased* estimator, *marginal* $M_{jj}$ corresponding to the $j^{\text{th}}$ signal coefficient and given by:

$$M_{jj} = \frac{1}{m} \sum_{i=1}^{m} y_i^2 a_{ij}^2, \qquad \text{for} \qquad j \in \{1, \ldots n\}. \tag{2}$$

The marginals themselves do not directly produce signal coefficients, but the "weight" of each marginal identifies the true signal support. Then, a spectral technique based on [13, 23, 22] constructs an initial estimate $\mathbf{x^0}$. To accurately estimate support, earlier works [13, 23] assume that the magnitudes of the nonzero signal coefficients are all sufficiently large, i.e., $\Omega\left(\|\mathbf{x}^*\|_2 / \sqrt{s}\right)$, which can be unrealistic, violating the power-decay law. Our analysis resolves this issue by *relaxing* the requirement of accurately identifying the support, without any tuning parameters, unlike [22]. We claim that a coarse estimate of the support is good enough, since the errors would correspond to small coefficients. Such "noise" in the signal estimate can be controlled with a sufficient number of samples. Instead, we show that a simple pruning step that rejects the smallest $n - k$ coordinates, followed by the spectral procedure of [23], gives us the initialization that we need. Concretely, if elements of $\mathbf{A}$ are distributed as per standard normal distribution $\mathcal{N}(0, 1)$, a weighted correlation matrix $\mathbf{M} = \frac{1}{m} \sum_{i=1}^{m} y_i^2 \mathbf{a}_i \mathbf{a}_i^{\top}$, can be constructed, having diagonal elements $M_{jj}$. Then, the diagonal elements of this expectation matrix $\mathbb{E}\left[\mathbf{M}\right]$ are given by:

$$\mathbb{E}\left[M_{jj}\right] = \|\mathbf{x}^*\|^2 + 2x_j^{*2} \tag{3}$$

exhibiting a clear separation when analyzed for $j \in S$ and $j \in S^c$. We can hence claim, that signal marginals at locations on the diagonal of $\mathbf{M}$ corresponding to $j \in S$ are larger, on an average, than those for $j \in S^c$. Based on this, we evaluate the diagonal elements $M_{jj}$ and reject $n - k$ coordinates corresponding to the smallest marginals obtain a crude approximation of signal support $\hat{S}$. Using a spectral technique, we find an initial vector in the reduced space, which is close to the true signal, if $m = \mathcal{O}\left(s^2 \log n\right)$.

**Theorem 3.1.** *The initial estimate* $\mathbf{x^0}$*, which is the output of Algorithm 1, is a small constant distance* $\delta_0$ *away from the true s-sparse signal* $\mathbf{x}^*$*, i.e.,*

$$\operatorname{dist}\left(\mathbf{x^0}, \mathbf{x}^*\right) \leq \delta_0 \|\mathbf{x}^*\|_2,$$

*where $0 < \delta_0 < 1$, as long as the number of (Gaussian) measurements satisfy, $m \geq Cs^2 \log mn$, with probability greater than $1 - \frac{8}{m}$.*

This theorem is proved via Lemmas C.1 through C.4 (Appendix C), and the argument proceeds as follows. We evaluate the marginals of the signal $M_{jj}$, in broadly two cases: $j \in S$ and $j \in S^c$. The key idea is to establish one of the following: (1) If the signal coefficients obey $\min_{j \in S} |x_j^*| = C \|x^*\|_2 /\sqrt{s}$, then, *w.h.p.* there exists a clear separation between the marginals $M_{jj}$ for $j \in S$ and $j \in S^c$. Then Algorithm 1 picks up the correct support (i.e. $\hat{S} = S$); (2) if there is no such restriction, even then the support picked up in Algorithm 1, $\hat{S}$, contains a bulk of the correct support $S$. The incorrect elements of $\hat{S}$ induce negligible error in estimating the intial vector. These approaches are illustrated in Figures 4 and 5 in Appendix C. The marginals $M_{jj} < \Theta$, *w.h.p.*, for $j \in S^c$ and $M_{jj} > \Theta$, $j \in S_+$, where $S_+$ is a big chunk of the picked support $S_+ \subseteq \hat{S}$, $S_+ = \{j \in S : x_j^{*2} \geq 15\sqrt{(\log mn)/m} \|x^*\|_2\}$ are separated by threshold $\Theta$ (Lemmas C.1 and C.2). The identification of the support $\hat{S}$ (which provably contains a significant chunk $S_+$ of the true support $S$) is used to construct the truncated correlation matrix $M_{\hat{S}}$. The top singular vector of this matrix $M_{\hat{S}}$, gives us a good initial estimate $x^0$.

The final step of Algorithm 1 requires a scaling of the normalized vector $v_1$ by a factor $\phi$, which conserves the power in the signal (Lemma F.1 in Appendix F), whp, where $\phi^2$ which is defined as

$$\phi^2 = \frac{1}{m} \sum_{i=1}^{m} y_i^2. \tag{4}$$

## 3.2 Descent to optimal solution

After obtaining an initial estimate $x^0$, we construct a method to accurately recover $x^*$. For this, we adapt the alternating minimization approach from [13]. The observation model (1) can be restated as:

$$\text{sign}\left(\langle a_i, x^* \rangle\right) \circ y_i = \langle a_i, x^* \rangle \quad \text{for} \quad i = \{1, 2, \dots m\}.$$

We introduce the *phase vector* $p \in \mathbb{R}^m$ containing (unknown) signs of measurements, i.e., $p_i = \text{sign}\left(\langle a_i, x \rangle\right), \; \forall i$ and *phase matrix* $P = \text{diag}(p)$. Then our measurement model gets modified as $P^* y = Ax^*$, where $P^*$ is the true phase matrix. We then minimize the loss function composed of variables $x$ and $P$,

$$\min_{\|x\|_0 \leq s, P \in \mathcal{P}} \|Ax - Py\|_2. \tag{5}$$

Here $\mathcal{P}$ is a set of all diagonal matrices $\in \mathbb{R}^{m \times m}$ with diagonal entries constrained to be in $\{-1, 1\}$. Hence the problem stated above is *not convex*. Instead, we alternate between estimating $P$ and $x$ as follows: (1) if we fix the signal estimate $x$, then the minimizer $P$ is given in closed form as $P = \text{diag}(\text{sign}(Ax))$; we call this the *phase estimation* step; (2) if we fix the phase matrix $P$, the sparse vector $x$ can be obtained by solving the *signal estimation* step:

$$\min_{x, \|x\|_0 \leq s} \|Ax - Py\|_2. \tag{6}$$

We employ the CoSaMP [41] algorithm to (approximately) solve the non-convex problem (6). We do not need to explicitly obtain the minimizer for (6) but only show a sufficient descent criterion, which we achieve by performing a careful analysis of the CoSaMP algorithm. For analysis reasons, we require that the entries of the input sensing matrix are distributed according to $\mathcal{N}(0, 1/\sqrt{m})$. This can be achieved by scaling down the inputs to CoSaMP: $A^t, P^{t+1}y$ by a factor of $\sqrt{m}$ (see $x$-update step of Algorithm 2). Another distinction is that we use a "warm start" CoSaMP routine for each iteration where the initial guess of the solution to (6) is given by the current signal estimate.

We now analyze our proposed descent scheme. We obtain the following theoretical result:

**Theorem 3.2.** *Given an initialization $x^0$ satisfying Algorithm 1, if we have number of (Gaussian) measurements $m \geq Cs \log \frac{n}{s}$, then the iterates of Algorithm 2 satisfy:*

$$\text{dist}\left(x^{t+1}, x^*\right) \leq \rho_0 \text{dist}\left(x^t, x^*\right). \tag{7}$$

*where $0 < \rho_0 < 1$ is a constant, with probability greater than $1 - e^{-\gamma m}$, for positive constant $\gamma$.*

The proof of this theorem can be found in Appendix E.

# 4 Block-sparse phase retrieval

The analysis of the proofs mentioned so far, as well as experimental results suggest that we can reduce sample complexity for successful sparse phase retrieval by exploiting further structural information about the signal. Block-sparse signals $\mathbf{x}^*$, can be said to be following a sparsity model $\mathcal{M}_{s,b}$, where $\mathcal{M}_{s,b}$ describes the set of all block-sparse signals with $s$ non-zeros being grouped into uniform predetermined blocks of size $b$, such that block-sparsity $k = \frac{s}{b}$. We use the index set $j_b = \{1, 2 \ldots k\}$, to denote block-indices. We introduce the concept of *block marginals*, a block-analogue to signal marginals, which can be analyzed to crudely estimate the block support of the signal in consideration. We use this formulation, along with the alternating minimization approach that uses model-based CoSaMP [30] to descend to the optimal solution.

## 4.1 Initialization

Analogous to the concept of marginals defined above, we introduce *block marginals* $M_{j_b j_b}$, where $M_{jj}$ is defined as in (2). For block index $j_b$, we define:

$$M_{j_b j_b} = \sqrt{\sum_{j \in j_b} M_{jj}^2}, \tag{8}$$

to develop the initialization stage of our *Block CoPRAM* algorithm. Similar to the proof approach of CoPRAM, we evaluate the block marginals, and use the top-$k$ such marginals to obtain a crude approximation $\hat{S}_b$ of the true block support $S_b$. This support can be used to construct the truncated correlation matrix $\mathbf{M}_{\hat{S}_b}$. The top singular vector of this matrix $\mathbf{M}_{\hat{S}_b}$ gives a good initial estimate $\mathbf{x}^0$ (Algorithm 3, Appendix A) for the Block CoPRAM algorithm (Algorithm 4, Appendix A). Through the evaluation of block marginals, we proceed to prove that the sample complexity required for a good initial estimate (and subsequently, successful signal recovery of block sparse signals) is given by $\mathcal{O}\left(ks \log n\right)$. This essentially reduces the sample complexity of signal recovery by a factor equal to the block-length $b$ over the sample complexity required for standard sparse phase retrieval.

**Theorem 4.1.** *The initial vector $\mathbf{x}^0$, which is the output of Algorithm 3, is a small constant distance $\delta_b$ away from the true signal $\mathbf{x}^* \in \mathcal{M}_{s,b}$, i.e.,*

$$\mathrm{dist}\left(\mathbf{x}^0, \mathbf{x}^*\right) \le \delta_b \left\|\mathbf{x}^*\right\|_2,$$

*where $0 < \delta_b < 1$, as long as the number of (Gaussian) measurements satisfy $m \ge C \frac{s^2}{b} \log mn$ with probability greater than $1 - \frac{8}{m}$.*

The proof can be found in Appendix D, and carries forward intuitively from the proof of the compressive phase-retrieval framework.

## 4.2 Descent to optimal solution

For the descent of Block CoPRAM to optimal solution, the phase-estimation step is the same as that in CoPRAM. For the signal estimation step, we attempt to solve the same minimization as in (6), except with the additional constraint that the signal $\mathbf{x}^*$ is *block sparse*,

$$\min_{\mathbf{x} \in \mathcal{M}_{s,b}} \|\mathbf{A}\mathbf{x} - \mathbf{P}\mathbf{y}\|_2, \tag{9}$$

where $\mathcal{M}_{s,b}$ describes the block sparsity model. In order to approximate the solution to (9), we use the *model-based CoSaMP* approach of [30]. This is a straightforward specialization of the CoSaMP algorithm and has been shown to achieve improved sample complexity over existing approaches for standard sparse recovery.

Similar to Theorem 3.2 above, we obtain the following result (the proof can be found in Appendix E):

**Theorem 4.2.** *Given an initialization $\mathbf{x}^0$ satisfying Algorithm 3, if we have number of (Gaussian) measurements $m \ge C \left(s + \frac{s}{b} \log \frac{n}{s}\right)$, then the iterates of Algorithm 4 satisfy:*

$$\mathrm{dist}\left(\mathbf{x}^{t+1}, \mathbf{x}^*\right) \le \rho_b \mathrm{dist}\left(\mathbf{x}^t, \mathbf{x}^*\right). \tag{10}$$

*where $0 < \rho_b < 1$ is a constant, with probability greater than $1 - e^{-\gamma m}$, for positive constant $\gamma$.*

The analysis so far has been made for uniform blocks of size $b$. However the same algorithm can be extended to the case of sparse signals with *non-uniform* blocks or clusters (refer Appendix A).

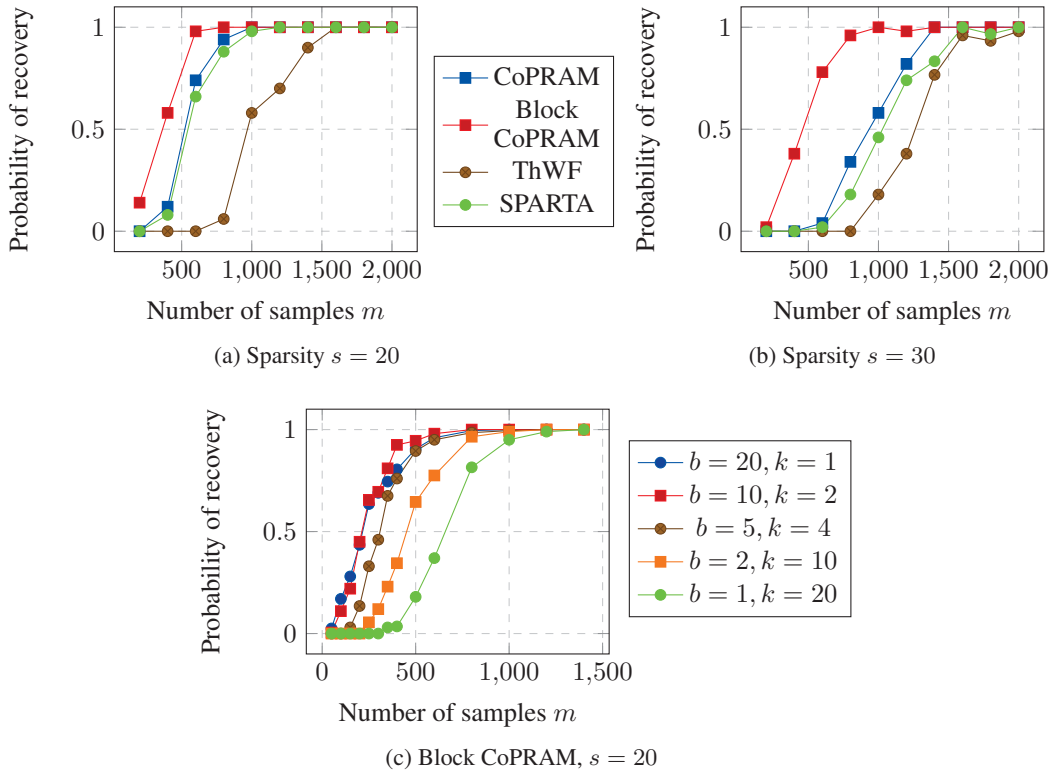

(a) Sparsity $s = 20$

(b) Sparsity $s = 30$

(c) Block CoPRAM, $s = 20$

Figure 1: Phase transitions for signal of length $n = 3,000$, sparsity $s$ and block length $b$ (a) $s = 20$, $b = 5$, (b) $s = 30$, $b = 5$, and (c) $s = 20$, $b = 20, 10, 5, 2, 1$ (Block CoPRAM only).

## 5 Experiments

We explore the performance of the CoPRAM and Block CoPRAM on synthetic data. All numerical experiments were conducted using MATLAB 2016a on a computer with an Intel Xeon CPU at 3.3GHz and 8GB RAM. The nonzero elements of the unit norm vector $\mathbf{x}^* \in \mathbb{R}^{3000}$ are generated from $\mathcal{N}(0, 1)$. We repeated each of the experiments (fixed $n, s, b, m$) in Figure 1 (a) and (b), for 50 and Figure 1 (c) for 200 independent Monte Carlo trials. For our simulations, we compared our algorithms CoPRAM and Block CoPRAM with Thresholded Wirtinger flow (Thresholded WF or ThWF) [22] and SPARTA [23]. The parameters for these algorithms were carefully chosen as per the description in their respective papers.

For the first experiment, we generated phase transition plots by evaluating the probability of empirical successful recovery, i.e. number of trials out of 50. The recovery probability for the four algorithms is displayed in Figure 1. It can be noted that increasing the sparsity of signal shifts the phase transitions to the right. However, the phase transition for Block CoPRAM has a less apparent shift (suggesting that sample complexity of $m$ has sub-quadratic dependence on $s$). We see that Block CoPRAM exhibits lowest sample complexity for the phase transitions in both cases (a) and (b) of Figure 1.

For the second experiment, we study the variation of phase transition with block length, for Block CoPRAM (Figure 1(c)). For this experiment we fixed a signal of length $n = 3,000$, sparsities $s = 20, k = 1$ for a block length of $b = 20$. We observe that the phase transitions improve with increase in block length. At block sparsity $\frac{s}{b} = \frac{20}{10} = 2$ (for large $b$, $b \to s$), we observe a saturation effect and the regime of the experiment is very close to the information theoretic limit.

Several additional phase transition diagrams can be found in Figure 2 in Appendix B. The running time of our algorithms compare favorably with Thresholded WF and SPARTA (see Table 2 in Appendix B). We also show that Block CoPRAM is more robust to noisy Gaussian measurements, in comparison to CoPRAM and SPARTA (see Figure 3 in Appendix B).

## Footnotes

[1]Exceptions to this rule are [24, 25, 26, 27, 28, 29] where very carefully crafted measurements $\mathbf{a}_i$ are used.

[2]We use the terms *sparse phase retrieval* and *compressive phase retrieval* interchangeably.

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
