[Supplementary Material]

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

* $\mathbf{P} = \text{diag}\left(\mathbf{p}\right)$. Then our measurement model gets modified as $\mathbf{P}^* \mathbf{y} = \mathbf{Ax}^*$, where $\mathbf{P}^*$ is the true phase matrix. We then minimize the loss function composed of variables $\mathbf{x}$ and $\mathbf{P}$,

$$\min_{\|\mathbf{x}\|_0 \leq s, \mathbf{P} \in \mathcal{P}} \|\mathbf{Ax} - \mathbf{Py}\|_2. \tag{5}$$

Here $\mathcal{P}$ is a set of all diagonal matrices $\in \mathbb{R}^{m \times m}$ with diagonal entries constrained to be in $\{-1, 1\}$. Hence the problem stated above is *not convex*. Instead, we alternate between estimating $\mathbf{P}$ and $\mathbf{x}$ as follows: (1) if we fix the signal estimate $\mathbf{x}$, then the minimizer $\mathbf{P}$ is given in closed form as $\mathbf{P} = \text{diag}\left(\text{sign}\left(\mathbf{Ax}\right)\right)$;

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

# A    Appendix - Block CoPRAM algorithm and extension

## A.1    Block CoPRAM algorithm

---

**Algorithm 3** Block CoPRAM: Initialization.

---

**input** $\mathbf{A}, \mathbf{y}, b, k$.

Compute signal power $\phi^2 = \frac{1}{m}\sum_{i=1}^{m} y_i^2$.

Compute block marginals $M_{j_b j_b} = \sqrt{\sum_{j \in j_b} M_{jj}^2}$    $\forall j_b$, where $M_{jj}$ is as in (2).

Select $\hat{S}_b \leftarrow j_b$'s corresponding to top-$k$ $M_{j_b j_b}$'s, $\hat{S}$ is signal support corresponding to blocks $\hat{S}_b$.

Compute $\mathbf{v}_1 \leftarrow$ top singular vector of $\mathbf{M}_{\hat{S}_b} = \frac{1}{m}\sum_{i=1}^{m} y_i^2 \mathbf{a}_{i\hat{S}} \mathbf{a}_{i\hat{S}}^{\top}$    $\in \mathbb{R}^{s \times s}$.

Compute $\mathbf{x}^0 \leftarrow \phi \mathbf{v}$ where $\mathbf{v} \leftarrow \mathbf{v}_1$ for $\hat{S}$, and $\mathbf{0} \in \mathbb{R}^{n-s}$ for $\hat{S}^c$.

**output** $\mathbf{x}^0$.

---

---

**Algorithm 4** Block CoPRAM: Descent.

---

**input** $\mathbf{A}, \mathbf{y}, \mathbf{x}^0, b, k, t_0$.

Initialize $\mathbf{x}^0$ according to Algorithm 3.

**for** $t = 0, \cdots, t_0 - 1$ **do**

$\mathbf{P}^{t+1} \leftarrow \mathrm{diag}\left(\mathrm{sign}\left(\mathbf{A}\mathbf{x}^t\right)\right)$.

$\mathbf{x}^{t+1} \leftarrow \mathrm{BlockCoSaMP}(\frac{1}{\sqrt{m}}\mathbf{A}, \frac{1}{\sqrt{m}}\mathbf{P}^{t+1}\mathbf{y}, b, k, \mathbf{x}^t)$.

**end for**

**output** $\mathbf{z} \leftarrow \mathbf{x}^{t_0}$.

---

## A.2    Extension to blocks of non-uniform sizes

The analysis so far has been made for uniform blocks of size $b$. However the same algorithm can be extended to the case of sparse signals with *non-uniform* blocks. Such a model is particularly useful for time-series signals where the nonzeros occur in "bursts" of variable lengths and start times.

Formally, consider the *clustered sparsity* model for 1D signals in $\mathbb{R}^n$, comprising signals with $s$ non-zeros that occur in no more than $k$ non-overlapping blocks (clusters), each of which exhibit potentially unknown sizes and locations. The above analysis does not immediately apply to this case; however, by the analysis approach of [37], we can show that any such clustered-sparse signal with parameters $(s, k)$ can be *simulated* using a *uniform* block-sparse signal with parameters $(s, 3k)$. Therefore, the only price to be paid is a tripling of the block sparsity parameter $k$. Provided we are willing to tolerate this increase, we can use exactly the same Block CoPRAM algorithm (including both the initialization as well as the descent stages) as described above, with only a constant factor increase in the sample complexity.

We note that this argument is only applicable to block-sparse 1D signals (such as time-domain signals); extending this argument to general clustered-sparse images and higher-dimensional data is much more involved, and we will not pursue this direction in this paper.

# B    Appendix - Additional experiments

We demonstrate the benefits of our algorithms CoPRAM and Block CoPRAM through an additional set of experiments and describe our previous experiments in further detail.

For Thresholded Wirtinger flow, we set parameters which were optimized based on a number of trial cases and were kept constant throughout all experiments, with values $\alpha = 1.5$, $\mu = 0.23$ and $\beta = 0.3$. Similarly, for SPARTA, we set the parameters to be $\gamma = 0.7$, $\mu = 1$ and $\mathrm{card}(\mathcal{I}_o) = \lceil \frac{m}{6} \rceil$ as mentioned in their paper. For our set of generated signals, the AltMinSparse method mentioned in [13] does not recover the signal in most cases (if the initialization stage fails to pick the correct support, the subsequent AltMinPhase procedure can never give a good solution). We therefore do not include this algorithm for comparisons. Figure 2 represents phase transition diagrams, where number of measurements $m$ is spanned from $m = 200$ to $m = 2000$ in steps of 200. Similarly

(a) Thresholded WF    (b) SPARTA    (c) CoPRAM    (d) Block CoPRAM

Figure 2: Phase transition plots for different algorithms, with signal length $n = 3000$, with uniform block length of $b = 5$.

Table 2: Mean running time of different algorithms at $s = 25$.

| Algorithm | CoPRAM | Block CoPRAM | SPARTA | Thresholded WF |
|---|---|---|---|---|
| $m$ at phase transition (suc freq = 1) | 1,600 | 1,400 | 1,800 | 2,000 |
| mean running time (s) | 0.4000 | 0.3258 | 0.3080 | 0.5808 |

signal sparsity $s$ is swept from $s = 5$ to $s = 50$ in steps of 5. The block lengths considered for all experiments in Figure 2 is $b = 5$. It can be noted that CoPRAM (1 (c)) and SPARTA (1 (b)) perform comparably, while Block CoPRAM (1 (d)) performs the best among all four algorithms, in terms of sample complexity. The mean running time of the algorithms for different algorithms is tabulated in Table 2. It can be noted that the running times of our algorithms CoPRAM and Block CoPRAM are at par with SPARTA and Thresholded WF.

***Effect of noise:*** For our third experiment, we study the effect of noise on the measurements of the form $y_i = |\langle \mathbf{a}_i, \mathbf{x}^* \rangle + e_i|$, for $i \in \{1, 2 \ldots m\}$. The noise vector $\mathbf{e} \in \mathbb{R}^m$ is sampled from a zero-mean Gaussian distribution $\mathcal{N}(0, \sigma^2)$, where $\sigma^2$ is determined using the input noise-signal-ratio (NSR). We compared CoPRAM, Block CoPRAM and SPARTA to analyze robustness to noisy measurements for amplitude only measurements (ThWF is excluded because they use quadratic measurements). We vary the input $\text{NSR} = \sigma^2 / \|\mathbf{x}^*\|_2^2$, from 0.1 to 1 in steps of 0.1. We fix signal parameters $n = 3,000, s = 20, b = 5, k = 4$ and number of measurements to $m = 1,600$. This experiment was run for 50 independent Monte Carlo trials. The variation of mean relative error $\|\mathbf{z} - \mathbf{x}^*\|_2 / \|\mathbf{x}^*\|_2$ (here $\mathbf{z} = \mathbf{x}^{t_0}$) can be seen in Figure 3. Block CoPRAM outperforms CoPRAM and SPARTA in all cases considered.

Figure 3: Variation of mean relative error in signal recovered v/s input NSR at $s = 20$ and $b = 5, k = 4$ for a signal of length $n = 3,000$, and number of measurements $m = 1,600$.

# C  Appendix - CoPRAM initialization

In this section we state the proofs related to the *initialization* in Algorithm 1, for compressive phase retrieval. This includes the proofs of Lemmas C.1 - C.4 which complete the proof of Theorem 3.1.

The outline of the proof is sketched out as follows. Using Lemma C.1, we can find an upper bound on marginals $M_{jj}$ for $j \in S$. Consequently,

$$\max_{j \in S^c} M_{jj} \leq \left(1 + 11\sqrt{\frac{\log mn}{m}}\right) \|\mathbf{x}^*\|_2^2 = \Theta_1 \tag{11}$$

with probability greater than $1 - \frac{5}{m}$. Marginals $M_{jj}$ for $j \in S$ can be evaluated in two ways:

1. Assuming a bound on the minimum element of $\mathbf{x}^*$: $x_{min}^{*2} \equiv \min_{j \in S} x_j^{*2} = \frac{C}{s}\|\mathbf{x}^*\|_2^2$. The proof then carries forward from the work in [23], where they arrive at the lower bound on the minimum marginal for $j \in S$, with probability greater than $1 - \frac{1}{m}$,

$$\min_{j \in S} M_{jj} \geq \|\mathbf{x}^*\|_2^2 + x_{min}^{*2} = \left(1 + \frac{C}{s}\right)\|\mathbf{x}^*\|_2^2 = \Theta_2,$$

given that $m \geq C_0 s^2 \log(mn)$. This proof is similar to that mentioned in Lemma C.2. Piecing these two together,

$$\min_{j \in S} M_{jj} \geq \left(1 + \frac{C}{s}\right)\|\mathbf{x}^*\|_2 > \left(1 + 11\sqrt{\frac{\log mn}{m}}\right)\|\mathbf{x}^*\|_2^2 \geq \max_{j \in S^c} M_{jj}. \tag{12}$$

which implies that the support picked up using the top $s$-marginals $M_{jj}$ is the true support with probability greater than $1 - \frac{6}{m}$, given $m \geq C_0 s^2 \log(mn)$, as long as there is a clear separation between $\Theta_1$ and $\Theta_2$ (i.e. $\Theta_1 < \Theta_2$). They proceed to show that with a high probability, $\|\mathbf{x}^0 - \mathbf{x}^*\|_2 \leq \delta_0 \|\mathbf{x}^*\|_2$, using Proposition 1 of [50], completing the proof of Theorem 3.1.

2. If there is no such assumption on the minimum entry $x_{min}^{*2}$, we proceed with a longer proof, as stated below using Lemmas C.2-C.4. The idea is to show that $\mathbf{x}^* \approx \mathbf{x}_{\hat{S}}^*$ and subsequently $\mathbf{x}_{\hat{S}}^* \approx \mathbf{x}^0$, effectively implying that $\mathbf{x}^0 \approx \mathbf{x}^*$.

This idea and the partition of support sets used in the proof have been illustrated in Figures 4 and 5.

Figure 4: Partition of supports considered for analysis of proof approach 1: assumption on $x_{min}^*$.

Figure 5: Partition of supports considered for analysis of proof approach 2.

**Lemma C.1.** *For all $j \in S^c$, with probability greater than $1 - \frac{5}{m}$, the corresponding marginals are upper-bounded as:*

$$M_{jj} \leq \left(1 + 11\sqrt{\frac{\log mn}{m}}\right) \|\mathbf{x}^*\|_2^2 = \Theta. \tag{13}$$

*Proof.* Evaluating the marginals:

$$M_{jj} - \phi^2 = \frac{1}{m}\sum_{i=1}^{m} y_i^2 \left(a_{ij}^2 - 1\right),$$

where $y_i$ is independent of $a_{ij}$ for all $j \in S^c$. Evaluating the tail bound in terms of a series of tail bounds for independent random variables $y_i$ and $a_{ij}$, one can use Lemma 4.1 of [58] for the $\chi_1^2$ variables $a_{ij}^2$ with weights $y_i^2$ (here $p \equiv n - s$):

$$\mathbb{P}\left[\sum_{i=1}^{m} y_i^2 \left(a_{ij}^2 - 1\right) > 2\sqrt{t}\left(\sum_{i=1}^{m} y_i^4\right)^{\frac{1}{2}} + 2\left(\max_i y_i^2\right) t\right] \leq \exp(-t) = \frac{1}{mp}.$$

Further, using the Chebyshev's inequality for $y_i^4$:

$$\mathbb{P}\left[\sum_{i=1}^{m} \frac{y_i^4}{\|\mathbf{x}^*\|_2^4} > 3m + \sqrt{96mt}\right] \leq \frac{1}{t^2} = \frac{1}{mp}.$$

Using the Gaussian tail bound for $y_i^2$ followed by union bound:

$$\mathbb{P}\left[\max_i \frac{y_i^2}{\|\mathbf{x}^*\|_2^2} > t\right] \leq 2m\exp\left(\frac{-t}{2}\right) = \frac{2}{mp^2} \leq \frac{2}{mp}.$$

With probability at most $\frac{4}{mp}$, for each $j \in S^c$, using a union bound on these three tail bounds,

$$\frac{1}{m}\sum_{i=1}^{m} y_i^2(a_{ij}^2 - 1) > 2\sqrt{3 + \sqrt{96p}}\,\|\mathbf{x}^*\|_2^2\,\sqrt{\frac{\log mp}{m}} + 8\,\|\mathbf{x}^*\|_2^2\,\frac{(\log mp)^2}{m}$$

$$> 2\sqrt{3 + \sqrt{96}}\,\|\mathbf{x}^*\|_2^2\,\sqrt{\frac{\log mp}{m}} + 8\,\|\mathbf{x}^*\|_2^2\,\frac{(\log mp)^2}{m}.$$

Using a union bound for all $j \in S^c$ ($p$ such), with probability at least $1 - \frac{4}{m}$,

$$\frac{1}{m}\sum_{i=1}^{m} y_i^2(a_{ij}^2 - 1) \leq 2\sqrt{3 + \sqrt{96}}\,\|\mathbf{x}^*\|_2^2\,\sqrt{\frac{\log mp}{m}} + 8\,\|\mathbf{x}^*\|_2^2\,\frac{(\log mp)^2}{m} \leq 8\sqrt{\frac{\log mp}{m}}\,\|\mathbf{x}^*\|_2^2. \tag{14}$$

Using Lemma F.1, for $m > C$, and using the fact that $p \leq n$:

$$M_{jj} = \frac{1}{m}\sum_{i=1}^{m} y_i^2 a_{ij}^2 \leq 8\sqrt{\frac{\log mn}{m}}\,\|\mathbf{x}^*\|_2^2 + \phi^2$$

$$M_{jj} \leq \left(1 + 11\sqrt{\frac{\log mn}{m}}\right)\|\mathbf{x}^*\|_2^2 = \Theta, \tag{15}$$

which establishes the upper bound on marginals associated with the zero-locations $j \in S^c$, with probability greater than $1 - \frac{5}{m}$. $\qquad\square$

**Lemma C.2.** *For $j \in S_+ \subseteq S$, with probability greater than $1 - \frac{2}{m}$, the corresponding marginals are lower-bounded as:*

$$M_{jj} \geq \left(1 + 11\sqrt{\frac{\log mn}{m}}\right)\|\mathbf{x}^*\|_2^2 = \Theta, \tag{16}$$

*where $S_+$ is defined as:*

$$S_+ = \left\{ j \in S \mid x_j^{*2} > 15\sqrt{\frac{\log mn}{m}} \, \|\mathbf{x}^*\|_2^2 \right\}. \tag{17}$$

*Subsequently, we can define $S_-$ as:*

$$S_- = \left\{ j \in S \mid x_j^{*2} \leq 15\sqrt{\frac{\log mn}{m}} \, \|\mathbf{x}^*\|_2^2 \right\}, \tag{18}$$

*with $S_+$ and $S_-$ forming a partition of $S$ and the corresponding energy in the elements $x_j, j \in S_-$ is lower-bounded as:*

$$\left\|\mathbf{x}_{S_-}^*\right\|_2^2 \leq 15\sqrt{\frac{s^2 \log mn}{m}} \, \|\mathbf{x}^*\|_2^2. \tag{19}$$

*Proof.* Evaluating the marginals:

$$M_{jj} - \phi^2 = \frac{1}{m}\sum_{i=1}^m y_i^2 \left(a_{ij}^2 - 1\right). \tag{20}$$

For $j \in S$, $y_i$ and $a_{ij}$ are dependent random variables. The marginal $M_{jj}$ can be evaluated through a concentration bounds on the two terms that compose the RHS of (20): $\frac{1}{m}\sum_{i=1}^m y_i^2 a_{ij}^2$ and $\frac{1}{m}\sum_{i=1}^m y_i^2$. This can be done by evaluating the expectation values:

$$\mathbb{E}\left[y_i^2\right] = \|\mathbf{x}^*\|_2^2$$
$$\mathbb{E}\left[y_i^2 a_{ij}^2\right] = \|\mathbf{x}^*\|_2^2 + 2x_j^{*2},$$
$$\mathbb{E}\left[y_i^4 a_{ij}^4\right] = 105 x_j^{*4} + 90 x_j^{*2}\left(\|\mathbf{x}^*\|_2^2 - x_j^{*2}\right) + 9\left(\|\mathbf{x}^*\|_2^2 - x_j^{*2}\right)^2.$$

Constructing variable $X_i = \|\mathbf{x}^*\|_2^2 + 2x_j^{*2} - y_i^2 a_{ij}^2$ which is upper bounded, with zero mean and bounded variance, we can use Lemma F.3 to establish a concentration bound with parameters:

$$X_i \leq \|\mathbf{x}^*\|_2^2 + 2x_j^{*2} \leq 3\|\mathbf{x}^*\|_2^2,$$
$$\mathbb{E}[X_i] = 0,$$
$$\mathbb{E}\left[X_i^2\right] = 20 x_j^{*4} + 68\|\mathbf{x}^*\|_2^2 x_j^{*2} + 8\|\mathbf{x}^*\|_2^4 \leq 96\|\mathbf{x}^*\|_2^4.$$

Using Lemma F.3, for each $j \in S$,

$$\mathbb{P}\left[\sum_{i=1}^m -X_i \leq -t\right] = \mathbb{P}\left[\sum_{i=1}^m y_i^2 a_{ij}^2 - m\left(\|\mathbf{x}^*\|_2^2 + 2x_j^{*2}\right) \leq -t\right]$$

$$\leq \exp\left(-\frac{t^2}{192\|\mathbf{x}^*\|_2^4 m}\right) \leq \frac{1}{mk} \tag{21}$$

This requires $t = \sqrt{192}\|\mathbf{x}^*\|_2^2\sqrt{m\log mk} \approx 13.86\|\mathbf{x}^*\|_2^2\sqrt{m\log mk} \leq 13.86\|\mathbf{x}^*\|_2^2\sqrt{m\log mn}$. This establishes the bound on the first term $\frac{1}{m}\sum_{i=1}^m y_i^2 a_{ij}^2$. Similarly, we can establish a bound on the second term $\frac{1}{m}\sum_{i=1}^m y_i^2$, which requires Lemma 4.1 of [58], with probability greater than $1 - \frac{1}{mk}$, for each $j \in S$:

$$\frac{1}{m}\sum_{i=1}^m y_i^2 - \|\mathbf{x}^*\|_2^2 \leq \left(2\sqrt{\frac{\log mk}{m}} + \frac{2\log mk}{m}\right)\|\mathbf{x}^*\|_2^2 \tag{22}$$

$$\leq 3\|\mathbf{x}^*\|_2^2\sqrt{\frac{\log mk}{m}} \leq 3\|\mathbf{x}^*\|_2^2\sqrt{\frac{\log mn}{m}} \tag{23}$$

for $m > C$. Combining these two concentration bounds (21), (22), taking a union bound for all $j \in S_+$ and substituting in (20):

$$M_{jj} - \phi^2 \geq 2x_j^{*2} - 17\sqrt{\frac{\log mn}{m}}\|\mathbf{x}^*\|_2^2, \tag{24}$$

which holds with probability at least $1 - \frac{2}{m}$.

If the set $S_+$, is constructed as in (17), then evaluating the bound in (24), we get:

$$M_{jj} - \phi^2 \geq 2x_j^{*2} - 17\sqrt{\frac{\log mn}{m}} \|\mathbf{x}^*\|_2^2$$

$$M_{jj} \geq \left(1 + 2x_j^{*2} - 19\sqrt{\frac{\log mn}{m}}\right) \|\mathbf{x}^*\|_2^2 \geq \left(1 + 11\sqrt{\frac{\log mn}{m}}\right) \|\mathbf{x}^*\|_2^2,$$

holds for all elements $j \in S_+$, with probability greater than $1 - \frac{2}{m}$, yielding the bound in (16). $\square$

**Lemma C.3.** *If $\hat{S}$ is chosen as in Algorithm 1, with probability greater than $1 - \frac{2}{m}$,*

$$\left\|\mathbf{x}^* - \mathbf{x}^*_{\hat{S}}\right\|_2 \leq \delta_1 \|\mathbf{x}^*\|_2, \tag{25}$$

*as long as the number of measurements $m$ follow the following bound*

$$m \geq Cs^2 \log mn. \tag{26}$$

*Proof.* If $\hat{S}$ is chosen such that it corresponds to the top-$s$ marginals $M_{jj}$, then it will pick up $S_+$ corresponding to large marginals $M_{jj} > \Theta$, $S_1 = S_- \cap \hat{S}$ and $S_2 = S^c \cap \hat{S}$ corresponding to small marginals $M_{jj} < \Theta$ ($S_+, S_1, S_2$ form a partition of $\hat{S}$ and $\mathrm{card}(\hat{S}) = s$, refer Figure 5 for illustration of the sets):

$$\mathbf{x}^*_{\hat{S}} = \mathbf{x}^*_{S_+} + \mathbf{x}^*_{S_1} + \mathbf{x}^*_{S_2}. \tag{27}$$

By definition $\mathbf{x}_{S^c} = \mathbf{0}$ and therefore $\mathbf{x}_{S_2} = \mathbf{0}$. If we can prove that $\mathbf{x}^* \approx \mathbf{x}^*_{\hat{S}}$ and $\mathbf{x}^*_{\hat{S}} \approx \mathbf{x}^0$, then we can claim that $\mathbf{x}^0 \approx \mathbf{x}^*$. First, we prove that $\left\|\mathbf{x}^* - \mathbf{x}^*_{\hat{S}}\right\|_2 \leq \delta_1 \|\mathbf{x}^*\|_2$:

$$\left\|\mathbf{x}^* - \mathbf{x}^*_{\hat{S}}\right\|_2^2 = \left\|\mathbf{x}^* - \mathbf{x}^*_{S_+} - \mathbf{x}^*_{S_1}\right\|_2^2 \leq \left\|\mathbf{x}^* - \mathbf{x}^*_{S_+}\right\|_2^2 + \left\|\mathbf{x}^*_{S_1}\right\|_2^2 \leq \left\|\mathbf{x}^* - \mathbf{x}^*_{S_+}\right\|_2^2 + \left\|\mathbf{x}^*_{S_-}\right\|_2^2.$$

By construction, $S_-$ and $S_+$ form a partion of $S$:

$$\mathbf{x}^* = \mathbf{x}^*_{S_-} + \mathbf{x}^*_{S_+},$$

$$\implies \left\|\mathbf{x}^* - \mathbf{x}^*_{\hat{S}}\right\|_2^2 \leq 2\left\|\mathbf{x}^*_{S_-}\right\|_2^2.$$

Using (19), we compute the bound,

$$\left\|\mathbf{x}^* - \mathbf{x}^*_{\hat{S}}\right\|_2^2 \leq 30\sqrt{\frac{s^2 \log mn}{m}} \|\mathbf{x}^*\|_2^2 \leq \delta_1^2 \|\mathbf{x}^*\|_2^2.$$

which is the required condition (25). This requires sample complexity $m$ to satisfy:

$$30\sqrt{\frac{s^2 \log mn}{m}} \leq \delta_1^2,$$

$$\implies m \geq \frac{900}{\delta_1^2} s^2 \log mn = C(\delta_1) s^2 \log mn. \tag{28}$$

$\square$

We have proved that $\mathbf{x}^* \approx \mathbf{x}^*_{\hat{S}}$. Now we need to prove that $\mathbf{x}^*_{\hat{S}} \approx \mathbf{x}^0$, which we do using Lemma C.4.

**Lemma C.4.** *With probability greater than $1 - \frac{8}{m}$*

$$\mathrm{dist}\left(\mathbf{x}^0, \mathbf{x}^*_{\hat{S}}\right) \equiv \min\left(\left\|\mathbf{x}^0 - \mathbf{x}^*_{\hat{S}}\right\|_2, \left\|\mathbf{x}^0 + \mathbf{x}^*_{\hat{S}}\right\|_2\right) \leq \delta_2 \|\mathbf{x}^*\|_2, \tag{29}$$

*as long as the number of measurements $m$ follow the following bound*

$$m \geq Cs \log n. \tag{30}$$

*Proof.* The top singular vector of $\mathbb{E}\left[\mathbf{M}\right]$ is equal to true $\mathbf{x}^*$:

$$\mathbb{E}\left[\mathbf{M}\right] = \mathbb{E}\left[\frac{1}{m}\sum_{j=1}^m y_j^2 \mathbf{a}_i \mathbf{a}_i^\top\right] = \left(\mathbf{I}_{n\times n} + 2\frac{\mathbf{x}^*}{\|\mathbf{x}^*\|_2}\frac{\mathbf{x}^{*\top}}{\|\mathbf{x}^*\|_2}\right)\|\mathbf{x}^*\|_2^2,$$

$$\text{similarly,} \quad \mathbb{E}\left[\mathbf{M}_S\right] = \mathbb{E}\left[\frac{1}{m}\sum_{i=1}^m y_i^2 \mathbf{a}_{iS} \mathbf{a}_{iS}^\top\right] = \left((\mathbf{I}_{n\times n})_S + 2\frac{\mathbf{x}^*}{\|\mathbf{x}^*\|_2}\frac{\mathbf{x}^{*\top}}{\|\mathbf{x}^*\|_2}\right)\|\mathbf{x}^*\|_2^2 = \mathbb{E}\left[\mathbf{M}\right].$$

We then define $\mathbf{M}_{\hat{S}} = \frac{1}{m}\sum_{i=1}^m y_i^2 \mathbf{a}_{i\hat{S}} \mathbf{a}_{i\hat{S}}^\top$ and $\mathbf{x^0}$ is the top singular vector of $\mathbf{M}_{\hat{S}}$.

Defining $S_3 \equiv (S \cup S_2) \subset (S \cup \hat{S})$, where $S_2 = \hat{S} \cap S^c$, then, $\text{card}(S_3) \leq 2s$, and,

$$\mathbb{E}\left[\mathbf{M}_{S_3}\right] = \mathbb{E}\left[\frac{1}{m}\sum_{i=1}^m y_i^2 \mathbf{a}_{iS_3} \mathbf{a}_{iS_3}^\top\right] = \left((\mathbf{I}_{n\times n})_{S_3} + 2\frac{\mathbf{x}^*}{\|\mathbf{x}^*\|_2}\frac{\mathbf{x}^{*\top}}{\|\mathbf{x}^*\|_2}\right)\|\mathbf{x}^*\|_2^2.$$

At this stage, we can invoke the proof idea from [22], as stated in Lemma F.2 from Appendix F, to give the following bound,

$$\|\mathbf{M}_{S_3} - \mathbb{E}\left[\mathbf{M}_{S_3}\right]\|_2 \leq \delta \|\mathbf{x}^*\|_2^2$$

with probability at least $1 - \frac{1}{m}$, as long as $m \geq Cs\log n$. Now we can use the fact that $\hat{S} \subset S_3$, so that,

$$\left\|\mathbf{M}_{\hat{S}} - \mathbb{E}\left[\mathbf{M}_{\hat{S}}\right]\right\|_2 \leq \|\mathbf{M}_{S_3} - \mathbb{E}\left[\mathbf{M}_{S_3}\right]\|_2 \leq \delta \|\mathbf{x}^*\|_2^2.$$

Since $\mathbf{M}_{\hat{S}}$ can be seen as a perturbation of $\mathbb{E}\left[\mathbf{M}_{\hat{S}}\right]$, where the top two singular values of $\mathbb{E}\left[\mathbf{M}_{\hat{S}}\right]$ are spaced $2\left\|\mathbf{x}^*_{\hat{S}}\right\|_2^2$ apart, we can use the Sin-Theta theorem [59] to bound the difference between the normalized top-singular vectors $\mathbf{x^0}$ of $\mathbf{M}_{\hat{S}}$ and $\mathbf{x}_{\hat{S}}$ of $\mathbb{E}\left[\mathbf{M}_{\hat{S}}\right]$ as,

$$\text{dist}\left(\mathbf{x^0}, \mathbf{x}^*_{\hat{S}}\right) \leq \frac{\delta \|\mathbf{x}^*\|_2^2}{2\|\mathbf{x}^*\|_2^2} = \frac{\delta}{2}$$

$$\implies \min\left(\|\mathbf{x^0} - \mathbf{x}^*_{\hat{S}}\|_2, \|\mathbf{x^0} + \mathbf{x}^*_{\hat{S}}\|_2\right) \leq \sqrt{2\left(1 - \sqrt{1 - \frac{\delta^2}{4}}\right)} \leq \delta_2$$

Hence, with probability greater than $1 - \frac{8}{m}$, Lemma C.4 holds. □

Combining Lemmas C.3 and C.4, we have the final result:

$$\text{dist}\left(\mathbf{x^0}, \mathbf{x}^*\right) = \min\left(\left\|\mathbf{x^0} - \mathbf{x}^*\right\|_2, \left\|\mathbf{x^0} + \mathbf{x}^*\right\|_2\right) \leq \delta_0 \|\mathbf{x}^*\|_2$$

as long as the number of measurements $m$ follow the bound in (26). Hence the initial vector $\mathbf{x^0}$ is upto a constant factor away from the true vector $\mathbf{x}^*$. The constant $\delta_0 \leq \delta_1 + \delta_2$ can be decreased by increasing the number of samples (see equation (28)). This completes the proof of Theorem 3.1.

# D   Appendix - Block CoPRAM initialization

In this section we state the proofs related to the *initialization* for Block CoPRAM in Algorithm 3, for block sparse signals.

We prove theorem 4.1 for the initialization stage of Block CoPRAM as follows.

**Theorem 4.1.** *The initial vector $\mathbf{x^0}$, which is the output of Algorithm 3, is a small constant distance $\delta_b$ away from the true signal $\mathbf{x}^* \in \mathcal{M}_{s,b}$, i.e.,*

$$\text{dist}\left(\mathbf{x^0}, \mathbf{x}^*\right) \leq \delta_b \|\mathbf{x}^*\|_2,$$

*where $0 < \delta_b < 1$, as long as the number of (Gaussian) measurements satisfy $m \geq C\frac{s^2}{b}\log mn$ with probability greater than $1 - \frac{8}{m}$.*

*Proof.* Evaluating the marginals $M_{j_b j_b}$, for all $j_b \in S_b^c$, from (11), with probability greater than $1 - \frac{5}{m}$, we have:

$$M_{j_b j_b} \leq \left( 1 + 11 \sqrt{\frac{\log mn}{m}} \right) \sqrt{b} \, \|\mathbf{x}^*\|_2^2. \tag{31}$$

Evaluating the block marginals $M_{j_b j_b}$, for $j_b \in S_b$, we use a modification of (21), with probability less than $\exp\left( -\frac{mt^2}{192\|\mathbf{x}^*\|_2^4} \right) \leq \frac{1}{mn}$,

$$\frac{1}{m} \sum_{i=1}^{m} -X_i \leq -t$$

$$\frac{1}{m} \sum_{i=1}^{m} y_i^2 a_{ij}^2 - \left( \|\mathbf{x}^*\|_2^2 + 2x_j^{*2} \right) \leq -t$$

Rearranging the terms,

$$\sum_{j \in j_b} M_{jj}^2 \leq \sum_{j \in j_b} \left[ \left( \|\mathbf{x}^*\|_2^2 - t \right) + 2x_j^{*2} \right]^2,$$

$$\leq b \left( \|\mathbf{x}^*\|_2^2 - t \right)^2 + 4 \left\| \mathbf{x}_{j_b}^* \right\|_2^4 + 4\sqrt{b} \left\| \mathbf{x}_{j_b}^* \right\|_2^2 \left( \|\mathbf{x}^*\|_2^2 - t \right),$$

$$\implies M_{j_b j_b} \leq \sqrt{b} \left( \|\mathbf{x}^*\|_2^2 - t \right) + 2 \left\| \mathbf{x}_{j_b}^* \right\|_2^2.$$

where the final expression holds with probability less than $\frac{b}{mn}$. Here, we have used he shorthand $\left\| \mathbf{x}_{j_b}^* \right\|_2^2 \equiv \sum_{j \in j_b} x_j^{*2}$. Finally, taking a minimum over all such block marginals $j_b \in S_b$, with probability greater than $1 - \frac{1}{m}$,

$$M_{j_b j_b} \geq \sqrt{b} \left( \|\mathbf{x}^*\|_2^2 - t \right) + 2 \left\| \mathbf{x}_{j_b}^* \right\|_2^2,$$

$$\geq \sqrt{b} \|\mathbf{x}^*\|_2^2 + \left\| \mathbf{x}_{b_{min}}^* \right\|_2^2,$$

if $\sqrt{b}t = \left\| \mathbf{x}_{b_{min}}^* \right\|_2^2 \equiv \min_{j_b \in S_b} \left\| \mathbf{x}_{j_b}^* \right\|_2^2$. Assuming that $\left\| \mathbf{x}_{b_{min}^*}^* \right\|_2^2 = \frac{C}{k} \|\mathbf{x}^*\|_2^2$, the following holds

$$\min_{j_b \in S_b} M_{j_b j_b} \geq \left( 1 + \frac{C}{\sqrt{b}k} \right) \sqrt{b} \|\mathbf{x}^*\|_2^2. \tag{32}$$

Equating the expression for probability,

$$m \geq 192 \frac{\|\mathbf{x}^*\|_2^4}{t^2} \log mn,$$

$$\geq Cbk^2 \log mn = C \frac{s^2}{b} \log mn,$$

which puts a bound on the block marginals for $j_b \in S_b$.

Hence, as long as $m \geq C \frac{s^2}{b} \log n$, there is a clear separation in the marginals, using (32) and (31),

$$\min_{j_b \in S_b} M_{j_b j_b} \geq \left( 1 + \frac{C}{\sqrt{b}k} \right) \sqrt{b} \|\mathbf{x}^*\|_2^2,$$

$$> \left( 1 + 11 \sqrt{\frac{\log mn}{m}} \right) \sqrt{b} \|\mathbf{x}^*\|_2^2,$$

$$\geq \max_{j_b \in S_b^c} M_{j_b j_b},$$

where $C$ is large enough. Given that there is a clear separation in the marginals, the block support $\hat{S}_b$ as picked up as in Algorithm 3, is exactly the true block support $S_b$.

It is then straightforward to show that the top singular vector of the truncated covariance matrix $\mathbf{M}_{\hat{S}_b}$ is actually close to the true block sparse vector $\mathbf{x}^*$, which holds with probability greater than $1 - \frac{1}{m}$.

Thus far, the proof requires an assumption on $\left\| \mathbf{x}^*_{b_{min}} \right\|_2$. We do away with this assumption as follows:

For evaluating block marginals $M_{j_b j_b}$ for $j_b \in S_b^c$, we can use the result of Lemma C.1, to obtain the same bound as in (31), with probability greater than $1 - \frac{5}{m}$,

$$M_{j_b j_b} \leq \left( 1 + 11 \sqrt{\frac{\log mn}{m}} \right) \sqrt{b} \left\| \mathbf{x}^* \right\|_2^2 .$$

For evaluating block marginals $M_{j_b j_b}$ for $j_b \in S_b$ we can use equations (17) and (18), and extend this model of signal supports to block supports defined as:

$$S_{b-} = \left\{ j_b \in S_b \mid \left\| \mathbf{x}^*_{j_b} \right\|_2^2 \equiv \sum_{j \in j_b} x_j^{*2} \leq 15 \sqrt{\frac{b \log mn}{m}} \left\| \mathbf{x}^* \right\|_2^2 \right\} ,$$

$$S_{b+} = \left\{ j_b \in S_b \mid \left\| \mathbf{x}^*_{j_b} \right\|_2^2 \equiv \sum_{j \in j_b} x_j^{*2} > 15 \sqrt{\frac{b \log mn}{m}} \left\| \mathbf{x}^* \right\|_2^2 \right\} .$$

Using equation (24), and LHS of (38),

$$M_{jj} \geq 2x_j^{*2} - 17 \sqrt{\frac{\log mn}{m}} \left\| \mathbf{x}^* \right\|_2^2 + \phi^2 ,$$

$$\geq 2x_j^{*2} + \left( 1 - 19 \sqrt{\frac{\log mn}{m}} \right) \left\| \mathbf{x}^* \right\|_2^2 .$$

Constructing block marginals as $M_{j_b j_b} \equiv \sqrt{\sum_{j \in j_b} M_{jj}^2}$,

$$M_{j_b j_b} \geq \sqrt{b} \left( 1 - 19 \sqrt{\frac{\log mn}{m}} \right) \left\| \mathbf{x}^* \right\|_2^2 + 2 \left\| \mathbf{x}^*_{j_b} \right\|_2^2 ,$$

$$\implies M_{j_b j_b} \geq \left( 1 + 11 \sqrt{\frac{b \log mn}{m}} \right) \left\| \mathbf{x}^* \right\|_2^2 .$$

We can then extend the proof of Lemma C.3 to give the partitions,

$$\mathbf{x}^*_{\hat{S}_b} = \mathbf{x}^*_{S_{b+}} + \mathbf{x}^*_{S_1} + \mathbf{x}^*_{S_2} ,$$

$$\mathbf{x}^* = \mathbf{x}^*_{S_{b-}} + \mathbf{x}^*_{S_{b+}} .$$

and the inequalities:

$$\left\| \mathbf{x}^* - \mathbf{x}^*_{\hat{S}_b} \right\|_2^2 \leq 2 \left\| \mathbf{x}^*_{S_{b-}} \right\|_2^2 ,$$

$$= 2 \sum_{j_b \in S_{b-}} \left\| \mathbf{x}^*_{j_b} \right\|_2^2 ,$$

$$\leq 15k \sqrt{\frac{b \log mn}{m}} \left\| \mathbf{x}^* \right\|_2^2 \leq \delta \left\| \mathbf{x}^* \right\|_2^2 .$$

This inequality gives us a bound on the number of measurements $m$, similar to (28),

$$m \geq \frac{15^2}{\delta^2} k^2 b \log mn = C(\delta) \frac{s^2}{b} \log mn ,$$

with probability greater than $1 - \frac{7}{m}$. This gives us the evaluation of block-marginals for $j_b \in S_b$ and $\hat{S}_b^c$, respectively. It is then straightforward to show that the top singular vector of the truncated covariance matrix $\mathbf{M}_{\hat{S}_b}$, given $\hat{S}_b$ is actually close to the true block sparse vector $\mathbf{x}^*$ with probability greater than $1 - \frac{1}{m}$. $\qquad \square$

# E    Appendix - CoPRAM and Block CoPRAM descent

In this section we state the proofs related to the *descent to optimal solution* in Algorithm 2 (CoPRAM), for sparse signals and Algorithm 4 (Block CoPRAM), for block sparse signals. This includes the proof of Theorem 3.2 and Theorem 4.2. We prove theorem 3.2 to show descent of the CoPRAM algorithm, as follows.

**Note:** For evaluation of the distance measure dist $(\cdot, \cdot)$, we only consider dist $(\mathbf{x}^t, \mathbf{x}^*) = \|\mathbf{x}^t - \mathbf{x}^*\|_2$, assuming that dist $(\mathbf{x^0}, \mathbf{x}^*) = \|\mathbf{x^0} - \mathbf{x}^*\|_2$ at the end of initialization stage. We claim that wlog, the same results would hold, if dist $(\mathbf{x^0}, \mathbf{x}^*) = \|\mathbf{x^0} + \mathbf{x}^*\|_2$.

**Theorem 3.2.** *Given an initialization $\mathbf{x^0}$ satisfying Algorithm 1, if we have number of (Gaussian) measurements $m \geq Cs \log \frac{n}{s}$, then the iterates of Algorithm 2 satisfy:*

$$\text{dist}\left(\mathbf{x}^{t+1}, \mathbf{x}^*\right) \leq \rho_0 \text{dist}\left(\mathbf{x}^t, \mathbf{x}^*\right). \tag{7}$$

*where $0 < \rho_0 < 1$ is a constant, with probability greater than $1 - e^{-\gamma m}$, for positive constant $\gamma$.*

---

**Algorithm 5** CoSaMP

---

**input** $\Phi = \frac{\mathbf{A}}{\sqrt{m}}, \mathbf{u} = \frac{\mathbf{P}^t \mathbf{y}}{\sqrt{m}}, s, \mathbf{x}^t$.
 1: Initialize

$$\mathbf{x}^{t+1,0} \leftarrow \mathbf{x}^t \quad \text{initialize to best possible estimate}$$
$$\mathbf{r} \leftarrow \mathbf{u} \quad \text{residue}$$
$$l \leftarrow 0 \quad \text{CoSaMP internal counter}$$

 2: **while** halting condition not true, **do**
 3:

$$l \leftarrow l + 1$$
$$\mathbf{v} \leftarrow \Phi^\top \mathbf{r} \quad \text{signal proxy}$$
$$\Omega \leftarrow \text{supp}(\mathbf{v}_{2s})$$
$$\Gamma \leftarrow \Omega \cup \text{supp}(\mathbf{x}^{t+1,l-1})$$
$$\mathbf{w} \leftarrow \Phi_\Gamma^\dagger \mathbf{u} \quad \text{corresponding to } \Gamma, 0 \text{ elsewhere}$$
$$\mathbf{x}^{t+1,l} \leftarrow \text{Truncate to top } s \text{ values of } \mathbf{w}, \text{ call this support } \Gamma_s$$
$$\mathbf{r} \leftarrow \mathbf{u} - \Phi \mathbf{x}^{t+1,l}$$

 4: **end while**
 5: $\mathbf{x}^{t+1,L} \leftarrow \Phi_{\Gamma_s}^\dagger u$.
**output** $\mathbf{x}^{t+1} \leftarrow \mathbf{x}^{t+1,L}$

---

To show the descent of our alternating minimization algorithm using CoSaMP, we need to analyze the reduction in error, per step of CoSaMP, (refer Algorithm 5) first:

$$\begin{aligned}
\left\|\mathbf{x}^{t+1,l+1} - \mathbf{x}^*\right\|_2 &= \left\|\mathbf{x}^{t+1,l+1} - \mathbf{w} + \mathbf{w} - \mathbf{x}^*\right\|_2, \\
&\leq 2 \left\|\mathbf{x}^* - \mathbf{w}\right\|_2
\end{aligned} \tag{33}$$

where $\mathbf{w}$ corresponds to the $\ell$'th run of CoSaMP for the $(t+1)^{th}$ update of $\mathbf{x}$. Using RIP of $\Phi = \frac{\mathbf{A}}{\sqrt{m}}$,

$$\left\|\mathbf{x}^{t+1,l+1} - \mathbf{x}^*\right\|_2 \leq \frac{2}{\sqrt{1 - \delta_s}} \left\|\Phi \mathbf{x}^* - \Phi \mathbf{w}\right\|_2, \tag{34}$$

with high probability, where $\delta_s$ is the RIP constant. Now, analyzing the inputs to CoSaMP, in the **x**-update step of Algorithm 2,

$$\mathbf{u} = \frac{\mathbf{P}^t \mathbf{y}}{\sqrt{m}},$$

$$= \text{sign}\left(\mathbf{A}\mathbf{x}^t\right) \circ \frac{|\mathbf{A}\mathbf{x}^*|}{\sqrt{m}},$$

$$= \text{sign}\left(\Phi\mathbf{x}^t\right) \circ \left\{\left(\Phi\mathbf{x}^*\right) \circ \text{sign}\left(\Phi\mathbf{x}^*\right)\right\},$$

$$= \Phi\mathbf{x}^* + \left(\text{sign}\left(\Phi\mathbf{x}^t\right)\text{sign}\left(\Phi\mathbf{x}^*\right) - \mathbf{1}\right) \circ \Phi\mathbf{x}^*,$$

$$\implies \mathbf{u} - \Phi\mathbf{x}^* = \left(\text{sign}\left(\Phi\mathbf{x}^t\right)\text{sign}\left(\Phi\mathbf{x}^*\right) - \mathbf{1}\right) \circ \Phi\mathbf{x}^*, \tag{35}$$

$$= E_{ph},$$

where $E_{ph} \equiv \left(\text{sign}\left(\Phi\mathbf{x}^t\right)\text{sign}\left(\Phi\mathbf{x}^*\right) - \mathbf{1}\right) \circ \Phi\mathbf{x}^*$, is error due to failure in estimating the correct phase.

Using equation (35) and substituting into equation (34), the per-step reduction in error for each run of CoSaMP is:

$$\left\|\mathbf{x}^{t+1,l+1} - \mathbf{x}^*\right\|_2 \leq \frac{2}{\sqrt{1-\delta_s}}\left\|\mathbf{u} - E_{ph} - \Phi\mathbf{w}\right\|_2$$

$$\leq \frac{2}{\sqrt{1-\delta_s}}\left\|\mathbf{u} - \Phi\mathbf{w}\right\|_2 + \frac{2}{\sqrt{1-\delta_s}}\left\|E_{ph}\right\|_2$$

$$\leq \frac{2}{\sqrt{1-\delta_s}}\left\|\mathbf{u} - \Phi_\Gamma \mathbf{w}_\Gamma\right\|_2 + \frac{2}{\sqrt{1-\delta_s}}\left\|E_{ph}\right\|_2$$

$$\leq \frac{2}{\sqrt{1-\delta_s}}\left\|\mathbf{u} - \Phi_\Gamma \mathbf{x}_\Gamma^*\right\|_2 + \frac{2}{\sqrt{1-\delta_s}}\left\|E_{ph}\right\|_2$$

$$\leq \frac{2}{\sqrt{1-\delta_s}}\left\|\Phi\mathbf{x}^* + E_{ph} - \Phi_\Gamma \mathbf{x}_\Gamma^*\right\|_2 + \frac{2}{\sqrt{1-\delta_s}}\left\|E_{ph}\right\|_2$$

$$\leq \frac{2}{\sqrt{1-\delta_s}}\left\|\Phi\mathbf{x}^* - \Phi_\Gamma \mathbf{x}_\Gamma^*\right\|_2 + \frac{4}{\sqrt{1-\delta_s}}\left\|E_{ph}\right\|_2$$

$$\leq \frac{2}{\sqrt{1-\delta_s}}\left\|\Phi_{\Gamma^c} \mathbf{x}_{\Gamma^c}^*\right\|_2 + \frac{4}{\sqrt{1-\delta_s}}\left\|E_{ph}\right\|_2$$

$$\leq 2\sqrt{\frac{1+\delta_s}{1-\delta_s}}\left\|\left(\mathbf{x}^* - \mathbf{x}^{t+1,l}\right)_{\Gamma^c}\right\|_2 + \frac{4}{\sqrt{1-\delta_s}}\left\|E_{ph}\right\|_2$$

$$= \rho_1 \left\|\left(\mathbf{x}^* - \mathbf{x}^{t+1,l}\right)_{\Gamma^c}\right\|_2 + \rho_2 \left\|E_{ph}\right\|_2,$$

where the first step is from using triangle inequality, the second step is from using the fact that **w** is exactly $3s$-sparse with support $\Gamma$. The third step is using the fact that truncation of **w** in $\Gamma, \in \mathbb{R}^{3s}$, is the minimizer of the LS problem $\text{argmin}_{\mathbf{x} \in \mathbb{R}^{3s}}\left\|\Phi_\Gamma \mathbf{x} - \mathbf{u}\right\|_2$, the fourth step uses (35) again, the final step uses RIP again (which holds with probability greater than $1 - e^{-\gamma_1 m}$, with $\gamma_1$ being a positive constant).

Finally, the first term in the previous inequality can be bounded using (Lemma 4.2 of CoSaMP [41], refer Lemma F.4), to yeild,

$$\left\|\mathbf{x}^{t+1,l+1} - \mathbf{x}^*\right\|_2 \leq \rho_1\rho_3 \left\|\mathbf{x}^* - \mathbf{x}^{t+1,l}\right\|_2 + \left(\rho_1\rho_4 + \rho_2\right)\left\|E_{ph}\right\|_2,$$

where $\rho_3, \rho_4$ are as stated in Lemma F.4. Assuming that CoSaMP is let to run a maximum of $L$ iterations,

$$\left\|\mathbf{x}^{t+1} - \mathbf{x}^*\right\|_2 \leq (\rho_1\rho_3)^L \left\|\mathbf{x}^* - \mathbf{x}^t\right\|_2 + \left(\rho_1\rho_4 + \rho_2\right)\left(1 + \rho_1\rho_3 + (\rho_1\rho_3)^2 \ldots (\rho_1\rho_3)^{L-1}\right)\left\|E_{ph}\right\|_2,$$

$$\leq (\rho_1\rho_3)^L \left\|\mathbf{x}^* - \mathbf{x}^t\right\|_2 + \frac{(\rho_1\rho_4 + \rho_2)}{(1 - \rho_1\rho_3)}\left\|E_{ph}\right\|_2. \tag{36}$$

The second part of this proof requires a bound on the phase error term $\|E_{ph}\|_2$:

$$E_{ph} = \pm \left(\text{sign}\left(\Phi\mathbf{x}^t\right) - \text{sign}\left(\Phi\mathbf{x}^*\right)\right) \circ \Phi\mathbf{x}^*.$$

We proceed to finish this proof by invoking Lemma E.1.

**Lemma E.1.** *As long as the initial estimate is a small distance away from the true signal* $\mathbf{x}^*$,

$$\text{dist}\left(\mathbf{x}^0, \mathbf{x}^*\right) \le \delta_0 \|\mathbf{x}^*\|_2,$$

*and subsequently,*

$$\text{dist}\left(\mathbf{x}^t, \mathbf{x}^*\right) \le \delta_0 \|\mathbf{x}^*\|_2,$$

*then the following bound holds,*

$$\frac{2}{m}\sum_{i=1}^{m}\left|\mathbf{a}_i^\top\mathbf{x}^*\right|^2 \mathbb{1}_{\{(\mathbf{a}_i^\top\mathbf{x}^t)(\mathbf{a}_i^\top\mathbf{x}^*)\le 0\}} \le \frac{2}{(1-\delta_0)^2}\left(\delta + \sqrt{\frac{21}{20}}\delta_0\right)\left\|\mathbf{x}^t - \mathbf{x}^*\right\|_2^2.$$

*with probability greater than* $1 - e^{-\gamma_2 m}$, *where* $\gamma_2$ *is a positive constant, as long as* $m > Cs\log\frac{n}{s}$. *We can use this to bound the phase error as,*

$$\|E_{ph}\|_2 \le \rho_5 \left\|\mathbf{x}^t - \mathbf{x}^*\right\|_2,$$

*where* $\rho_5 = \frac{\sqrt{2}}{(1-\delta_0)}\sqrt{\delta + \sqrt{\frac{21}{20}}\delta_0}$, $\delta \approx 0.001$.

This proof has been adapted from Lemma 7.19 of [42] and uses the generic chaining techniques of [43, 44]. Using this in addition to equation (36), we have our final per-step error reduction for a single run of CoPRAM (Algorithm 2), as:

$$\begin{aligned}\left\|\mathbf{x}^{t+1} - \mathbf{x}^*\right\|_2 &\le \left((\rho_1\rho_3)^L + \rho_5\frac{(\rho_1\rho_4 + \rho_2)}{(1 - \rho_1\rho_3)}\right)\left\|\mathbf{x}^t - \mathbf{x}^*\right\|_2,\\ &\le \rho_0\left\|\mathbf{x}^t - \mathbf{x}^*\right\|_2,\end{aligned} \tag{37}$$

where $\rho_0 < 1$.

**Evaluating convergence parameter** $\rho_0$**:.** To obtain per-step reduction in error, we require $\rho_0 < 1$. For sake of numerical analysis, $\delta_s, \delta_{2s}, \delta_{4s} \le 0.0001$, then $\rho_1 \approx 1$, $\rho_3 \approx 0.0002$. Let $\delta_0 = 0.012$, then $\rho_5 \approx 0.16$. Similarly, $\rho_2 \approx 4$ and $\rho_4 \approx 2$. Suppose CoSaMP is allowed to run for $L = 5$ iterations then, $\rho_0 \approx 0.96 < 1$.

The inequalities used for CoSaMP, particularly (33) can be made tighter, which would give less tight restrictions on the factor $\delta_0$, that controls how close the intial estimate is to the true signal $\mathbf{x}^*$.

We now restate theorem 4.2 for Block CoPRAM as follows.

**Theorem 4.2.** *Given an initialization* $\mathbf{x}^0$ *satisfying Algorithm 3, if we have number of (Gaussian) measurements* $m \ge C\left(s + \frac{s}{b}\log\frac{n}{s}\right)$, *then the iterates of Algorithm 4 satisfy:*

$$\text{dist}\left(\mathbf{x}^{t+1}, \mathbf{x}^*\right) \le \rho_b\text{dist}\left(\mathbf{x}^t, \mathbf{x}^*\right). \tag{10}$$

*where* $0 < \rho_b < 1$ *is a constant, with probability greater than* $1 - e^{-\gamma m}$, *for positive constant* $\gamma$.

The proof for this is a natural extention to the one we have proved in Theorem 3.2, and would use the results from the paper on model-based compressive sensing [30], wherever Block CoSaMP is invoked.

## F   Supplementary appendix

In this section we state some of the lemmas with or without proofs, used in Appendices C and E.

**Lemma F.1.** *With probability of at least* $1 - \frac{1}{m}$,

$$\left(1 - 2\sqrt{\frac{\log m}{m}}\right)\|\mathbf{x}^*\|_2^2 \leq \phi^2 \leq \left(1 + 3\sqrt{\frac{\log m}{m}}\right)\|\mathbf{x}^*\|_2^2. \tag{38}$$

*Proof.* Rotational invariance property of Gaussian distributions imply that $\mathbf{y}_i^2 \equiv (\sum_{j=1}^n a_{ij} x_j^*)^2$ has the same distrubution as $a_{ij}^2 \|\mathbf{x}^*\|_2^2$. Using Lemma 4.1 of [58] on $a_{ij}^2$, we can obtain the upper bound,

$$\mathbb{P}\left[\frac{1}{m}\sum_{i=1}^m a_{ij}^2 - 1 \geq 2\frac{\sqrt{m \log m}}{m} + 2\frac{\log m}{m}\right] \leq \exp\left(-\log m\right) = \frac{1}{m}.$$

Similarly, we can obtain the lower bound,

$$\mathbb{P}\left[\frac{1}{m}\sum_{i=1}^m a_{ij}^2 - 1 \leq -2\frac{\sqrt{m \log m}}{m}\right] \leq \exp\left(-\log m\right) = \frac{1}{m}.$$

The signal power $\phi^2$ is then bounded as

$$\left(1 - 2\sqrt{\frac{\log m}{m}}\right)\|\mathbf{x}^*\|_2^2 \leq \phi^2$$

$$\leq \left(1 + 2\sqrt{\frac{\log m}{m}} + 2\frac{\log m}{m}\right)\|\mathbf{x}^*\|_2^2$$

$$< \left(1 + 3\sqrt{\frac{\log m}{m}}\right)\|\mathbf{x}^*\|_2^2,$$

with probability at least $1 - \frac{1}{m}$, for $m > C$, large enough. If $m \approx 1000$, then the bounds are,

$$(1 - \delta)\|\mathbf{x}^*\|_2^2 \leq \phi^2 \leq (1 + \delta)\|\mathbf{x}^*\|_2^2,$$

where $\delta = 0.0207$. $\qquad\square$

**Lemma F.2.** *With probability at least* $1 - \frac{1}{m}$, *the following holds,*

$$\left\|\frac{1}{m}\sum_{i=1}^m \left|\mathbf{a}_{iS_3}^\top \mathbf{x}^*\right|^2 \mathbf{a}_{iS_3}\mathbf{a}_{iS_3}^\top - \left(\|\mathbf{x}^*\|_2^2 (\mathbf{I}_{n\times n})_{S_3} + 2\mathbf{x}^*\mathbf{x}^{*\top}\right)\right\|_2 \leq \delta\|\mathbf{x}^*\|_2^2$$

*where* $\mathrm{card}(S_3) \leq 2s$, *provided* $m > C(\delta)(2s)\log(2s)$.

This proof has been adapted from Lemma A.6 of [22].

**Lemma F.3.** *Suppose* $X_1 \ldots X_m$ *are i.i.d. centered, bounded real-valued random variables obeying*

$$X_i \leq b,$$
$$\mathbb{E}\left[X_i\right] = 0,$$
$$\mathbb{E}\left[X_i^2\right] = v^2,$$
$$\sigma^2 = \max\left\{b^2, v^2\right\},$$

*with cumulative distribution function of the standard normal distribution being denoted as*

$$\Phi(x) = \int_{-\infty}^x \phi(t)dt,$$

$$\phi(t) = \frac{1}{\sqrt{2\pi}}\exp\left(-\frac{t^2}{2}\right),$$

*then*

$$\mathbb{P}\left[\sum_{i=1}^{m} X_i \geq t\right] \leq \min\left\{\exp\left(-\frac{t^2}{2\sigma^2}\right), 25\left(1 - \Phi\left(\frac{t}{\sigma}\right)\right)\right\}.$$

*This establishes the tail probability of martingale with differences bounded from one side [60].*

**Lemma F.4.** *The $2s$-sparse residual error $\left\|(\mathbf{x}^* - \mathbf{x}^{t+1,l})_{\Gamma^c}\right\|_2$ can be upper bounded as,*

$$\left\|(\mathbf{x}^* - \mathbf{x}^{t+1,l})_{\Gamma^c}\right\|_2 \leq \left\|(\mathbf{x}^* - \mathbf{x}^{t+1,l})_{\Omega^c}\right\|_2 \leq \rho_3 \left\|(\mathbf{x}^* - \mathbf{x}^{t+1,l})\right\|_2 + \rho_4 \left\|E_{ph}\right\|_2$$

*where $\rho_3 = \frac{\delta_{2s} + \delta_{4s}}{1 - \delta_{2s}}$ and $\rho_4 = \frac{2\sqrt{1 + \delta_{2s}}}{1 - \delta_{2s}}$.*

This lemma has been adapted from Lemmas 4.2 and 4.3 of [41].