[Reviews · NeurIPS 2017]

Reviewer 1



The authors' rebuttal contains some points which should be explicitly included in a revised version if the paper is accepted. ----------- The authors study the problem of compressive (or sparse) phase retrieval, in which a sparse signal x\in\R^n is to be recovered from measurements abs(Ax), where A is a design (measurement) matrix, and abs() takes the entry-wise absolute value. The authors propose an iterative algorithm that applies CoSaMP in each iteration, and prove convergence to the correct, unknown sparse vector x with high probability, assuming that the number of measurements is at least C*s*log(n/s), where s is the sparsity of x and n is the ambient dimension. The paper is well written and clear, and the result is novel as far as I can tell. However, the problem under study is not adequately motivated. The authors only consider real-valued signals throughout the paper, yet bring motivation from real phase retrieval problems in science in which the signals are, without exception, complex. The authors aim to continue the work in refs. [21,22] which also considered only real-valued signals. Looking at those papers, I could find no satisfying motivation to study the problem as formulated. The algorithm proposed is compared with those proposed in refs. [21,22]. According to Table 1, it enjoys the same sample complexity requirement and the same running time (up to constants) as SPARTA [22] and TWF [21]. (While the sample complexity requirement is proved in the paper - Theorem 4.2 - I could not find a proof of the running time claimed in Table 1). As far as I could see, the only improvements of the proposed algorithm over prior art is the empirical performance (Section 6). In addition, the authors study a closely related problem, where the signal is assumed to be block sparse, and develop a version of the algorithm adapted to this problem. As for the sparse (non-block) version, in my opinion the limited simulation study offered by the authors is not sufficient to establish that the proposed algorithm improves in any way over those of [21,22]. Without a careful empirical study, or alternatively a formal results showing that the proposed algorithm improves over state of the art, the merit of the proposed algorithm is not sufficiently established. As for the block version, it seems that here the algorithm, which is specifically tailored to the block version, improves over state of the art. However, in my opinion the problem of recovery of *real* block-sparse signals from absolute value of linear measurements is not sufficiently motivated. Other comments: The authors assume a Gaussian measurement matrix. The main results (Theorem 4.1, Theorem 4.2, Theorem 5.2) must explicitly specify this assumption. As written now, the reader may incorrectly infer that the Theorem makes no assumptions on the measurement matrix.

Reviewer 2



This paper considers the phase retrieval problem under Gaussian random sampling with additional structural assumptions (e.g. sparsity and block sparsity of signals of interest). The authors propose a parameter-free nonconvex algorithm (by combining alternating minimization and CoSAMP), which allows to achieve appealing sample complexity and runtime all at once. Perhaps the most interesting message of this paper is the sample complexity for block-sparse signals. In particular, the sample complexity is O(s^2 / b) with b denoting block size and s the total sparsity. This seems to imply that the performance of sparse phase retrieval improves as block size increases (so that there is less uncertainty in support location). This is in stark contrast to sparse signal recovery case as changing the block size only affects the sample complexity by no more than a logarithmic factor. The paper is interesting and well-written. Below are some additional comments. 1. The thresholded WF has runtime O(n^2 log n), while CoPRAM has runtime O(s^2 n log n). So CoPRAM does not necessarily outperform threshold-WF, especially when s > sqrt{n}, right? This requires more discussion. 2. In Section 1.3, the authors mentioned that this work outperforms [22] as it allows to consider signals with power-law decay. If we use convex relaxation approaches (L1 + trace norm), the sample complexity for recovering power-law signals can be much better than O(s^2 log n); see [54] and [19]. Will CoPRAM also exhibit improved performance when restricted to signals with power-law decays? 3. Given that Section 1.3 entails detailed discussions of / comparisons with prior works, the authors may consider merging Section 1.3 and Section 2. 4. Section 4: the authors might need to be a bit careful when saying the problem "has multiple local minima". Given that the authors only consider i.i.d. Gaussian measurements and the sample size are quite large (much larger than information-theoretic limits), it's possible (although I am not sure) that we will have benign geometric structures with no spurious local critical points. 5. After the authors describe the second stage of their algorithm, it would be helpful to compare the computational bottleneck with threshold WF (in particular, why Alt-min + CoSAMP achieves better computational complexity than WF when s = o(\sqrt{n})). This would be helpful messages for the readers. Typos: --- abstract: arises from to structured sparsity models --> arises from structured sparsity models --- Page 1: ... of the light waves but not its phase --> ... of the light waves but not their phase --- Page 4: there has been no rigorous results --> there has been no rigorous result